# CLIP-Enhance: Improving CLIP Zero-Shot Classification via von Mises-Fisher Clustering

## Abstract

Contrastive language-image pre-training (CLIP) has revolutionized computer vision by integrating natural language understanding with image analysis, enabling zero-shot classification without prior training on specific classes. However, recent efforts to improve the performance of frozen CLIP models through prompt tuning and adapter mechanisms have introduced additional system complexity and training requirements, thus undermining CLIP's inherent efficiency in zero-shot knowledge transfer. In this paper, we propose to address two common challenges in zero-shot classification using CLIP: 1) the misalignment between textual and image embeddings, and 2) the long-tailed distribution of CLIP's training dataset. Our approach, CLIP-Enhance, is motivated by a re-interpretation of CLIP zero-shot classification as a clustering problem on a hypersphere using a von Mises-Fisher mixture model. Inspired by the DINO self-supervised learning framework, we optimize this mixture model to simultaneously improve the alignment of textual and image embeddings as well as represent data distribution disparities between training and evaluation datasets. Empirically, we show that jointly optimizing for both embedding alignment and concentration via self-supervised learning improves CLIP zero-shot classification significantly across multiple benchmark datasets. We also show empirically how CLIP-Enhance mitigates problems (1) and (2), as well as its robustness to limited data through a series of additional experiments.

## 1 Introduction

Advancements in neural network architectures (Dosovitskiy et al., 2020; He et al., 2015; Krizhevsky et al., 2012) and the availability of large scale datasets (e.g. Imagenet (Deng et al., 2009b), COCO (Lin et al., 2014), SA-1B (Kirillov et al., 2023), Laion (Schuhmann et al., 2022)) have produced extraordinary results on supervised vision understanding tasks with closed-set assumptions (Carion et al., 2020; Touvron et al., 2021; Kirillov et al., 2023; Girshick, 2015). Despite their success in these scenarios, such techniques are often less effective when encountering new, unseen classes, as they rely on annotated data and often lack the ability to attain general visual representations. Inspired by humans' ability to infer novel visual categories based on text descriptions, recent research focused on multi-modal representation learning has established new possibilities for zero-shot, multi-modal classification. In this context, zero-shot classification consists of leveraging an available multi-modal feature representation to perform closed-set classification on an unlabelled dataset for which textual descriptions of the classes is available.

In particular, vision-language pre-training frameworks, such as CLIP (Radford et al., 2021) and subsequent similar work (e.g. ALIGN (Jia et al., 2021), BLIP (Li et al., 2022b), Florence (Yuan et al., 2021)), have established themselves as methods for supervised visual representation learning. These methods encode both image and text data to the same embedding space under the assumption that pre-training on large-scale noisy image-caption (text) pairs allows models to learn diverse visual concepts that are easily transferred to downstream tasks. For example, for zero-shot classification, embeddings generated from text describing each novel class can then be compared to embeddings generated from the images that are being classified.

While models such as CLIP represent a significant advancement in multi-modal representation learning, using CLIP embeddings directly for zero-shot classification performs remarkably poorly on many simple datasets such as MNIST (see Table 1). Moreover, many methods designed to use

CLIP embeddings for few- or zero-shot classification make two tacit assumptions about the second central moments of image and text embedding distributions which are often violated in real-world applications (Zhang et al., 2021; Shu et al., 2022; Zhou et al., 2022c; Li et al., 2024; Guo et al., 2023). First, they assume that embeddings for images of a given class are distributed *anisotropically* around the canonical text embedding representing the reference text 'This is an image of a [CLASS].' In reality, the many-to-many nature of image captioning and the lack of aggregate or class-wide terms within the contrastive loss permits 'misalignment' between embeddings of text describing a given class and embeddings of images corresponding to that class. This manifests empirically as lower-than-desired cosine similarity values between image and text embeddings within a given class.

Second, there is an assumption that the sets of vectors representing different classes are spread across the representation space somewhat equally. In reality, CLIP's training dataset exhibits a long-tailed distribution, where certain images and captions occur much more frequently than others (e.g. cats vs. elephants) (Radford et al., 2021). This can produce different central second moments in embedding distributions for different classes (Wang & Isola, 2020). The distribution shift from a large scale training dataset to a specific downstream task can potentially exacerbate these differences and represents a challenge for zero-shot classification.

CLIP zero-shot classification has been interpreted as clustering (Radford et al., 2021) where the text embeddings form the cluster centers. To address the above issues of anisotropic and non-uniform embedding concentrations, this interpretation can be extended to consider clustering on a hyphersphere using a von Mises-Fisher (vMF) mixture model (Banerjee et al., 2005), which parameterizes each cluster by a mean direction vector and a concentration parameter. Leveraging the invariance of CLIP's representations, we jointly optimize both cluster mean directions and concentration parameters via a DINO-inspired (Caron et al., 2021) self-supervised learning (SSL) algorithm, which we call CLIP-Enhance (Fig. 1). Our approach also benefits from operating directly in text-prompt embedding space rather than in prompt space which allows it access to a set of finer resolution solutions than can be achieved via prompt engineering alone.

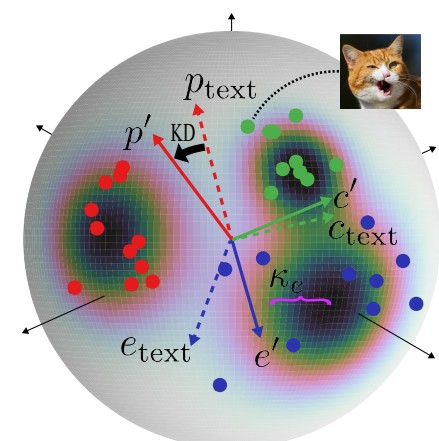

Figure 1: CLIP embedding hypersphere, showing data for "plane" (red), "cat" (green), and "elephant" (blue) classes. Note the initial misalignment between text embeddings (e.g. $p_{\text{text}}$, dashed lines) and the distribution means for the image embeddings (dots). After knowledge distillation, the new text embeddings (e.g. $p'$, solid lines) move towards the mean image embedding. Each class also has an embedding vector concentration $\kappa$ that CLIP-Enhance learns and which scales the final cosine similarity calculation non-linearly.

In summary, this paper presents the following contributions: (1) We re-formulate CLIP zero-shot classification as a vMF clustering problem on a hypersphere; and (2) we devise an SSL algorithm to jointly optimize both vision-text alignment in embedding space and embedding concentration estimates for the resulting vMF mixture model, addressing both anisotropic intra-class embedding distributions about the mean and non-uniform inter-class embedding concentrations. Together, this produces a state-of-the-art zero-shot classification system, which we show outperforms other methods on a variety of standard datasets. We also perform several ablation studies that show CLIP-Enhance performs well with limited compute and data resources, and the effect of system design choices.

## 2 RELATED WORK

Supervised pre-training on large-scale image classification tasks (e.g. ImageNet (Deng et al., 2009b)) and transferring to downstream tasks has been widely adopted (Simonyan & Zisserman, 2014; Girshick et al., 2014; He et al., 2022; Devlin et al., 2018; Dosovitskiy et al., 2020). However, constructing datasets at scale is challenging due to collection and labeling costs. Recently, vision-language pre-training (CLIP (Radford et al., 2021), ALIGN (Jia et al., 2021), BLIP (Li et al., 2022b), Florence (Yuan et al., 2021)) has emerged as a promising alternative for visual representation learning. Because the learned representations from CLIP and other models are general, additional work is often required to achieve state-of-the-art performance on specific downstream tasks. These efforts largely

fall into one of three categories depending on the compute and data resources available: fine-tuning models, few-shot learning, and zero-shot learning. Fine-tuning CLIP or similar models uses the general representations learned initially to jump-start learning patterns for a specific application (Wortsman et al., 2022; Han et al., 2024; Nam et al., 2024; Mao et al., 2024; Qiu et al., 2021). However, although fine-tuning is much more affordable than re-training large models entirely from scratch, it still requires substantial data and compute resources.

In an effort to more fruitfully exploit CLIP's low-cost transferability, some methods have attempted to leverage few-shot learning to improve CLIP's performance on specific datasets. These methods typically either focus on prompt tuning, where text inputs are learned or modified to produce text embeddings that better align with the corresponding image embeddings, or adapter methods, which learn a small adapter to modify the embeddings. For example, CoOp (Zhou et al., 2022d) proposes learning prompts for textual inputs to optimize classification in a few-shot setup, and CoCoOp (Zhou et al., 2022a) extends CoOp by adding context to prompt learning.

While CoOp and CoCoOp address the problem in prompt token space, a second class of approaches directly adapts the visual-text embedding space, allowing more fine-grained adjustments than prompt-based methods resulting in superior performance. CLIP-Adapter (Gao et al., 2024) appends an adapter module to produce adapted multi-modal features for both text and image modalities which are better aligned, and Tip-Adapter (Zhang et al., 2021) greatly reduces training cost by constructing a key-value cache model from the given set of labelled images in the few-shot setup. PromptKD (Li et al., 2024) also aims to learn prompts but additionally learns a non-linear image embedding projector to improve alignment by minimizing KL-divergence between student and teacher networks. However, the teacher model is trained initially in a supervised manner on few-shot, labelled data. Overall, these adaptation and prompt tuning methods often involve additional learnable parameters and still require extra training data, which undermine the core principle of CLIP's efficient zero-shot recognition.

In zero-shot learning, there have been few published approaches to improve CLIP's baseline performance. CALIP (Guo et al., 2023) proposes a parameter-free attention module, which guides visual and textual representations to interact with each other and explore cross-modal informative features via attention. CALIP blends image features with textual-aware signals and the textual features with visual-guided signal for better adaptive zero-shot classification. Although they both improve CLIP zero-shot, CLIP-Enhance and CALIP are not mutually exclusive and, in theory, may be used simultaneously. It is still an open question as to whether or not improvements in performance due to CALIP and CLIP-Enhance arise from the same underlying data instances.

The most directly related work to our own is TPT (Shu et al., 2022), however there are significant differences. First, TPT optimizes prompt embeddings by minimizing the entropy among predictions on augmented views of each individual test sample, whereas we optimize for the entire test dataset. This allows our method to leverage additional knowledge that TPT cannot fully exploit. Second, TPT relies on entropy minimization, while we frame the problem as knowledge distillation, which has been demonstrated to be more effective (Caron et al., 2021). Furthermore, we formulate the clustering task using von Mises-Fisher distributions, which accounts for the intra-class spread of embeddings, something other methods, including TPT, cannot do. Overall, CLIP-Enhance offers several benefits compared to other approaches in addition to higher accuracy, including the ability to adapt to the whole dataset at hand by addressing the inference problem directly and greater ease of use due to not having to fine-tune hyperparameters for each different dataset.

## 3 METHOD

In this section, we begin by establishing CLIP zero-shot classification as a clustering problem. Next, we introduce CLIP-Enhance and show how this clustering can be modeled as a von Mises-Fisher mixture model. Finally, we describe our self-supervised approach to distill and align the aforementioned vMF mixture model.

### 3.1 ZERO SHOT CLASSIFICATION USING CLIP

In zero-shot classification, the objective is to classify a set of novel images into a set of potentially unseen classes $C = \{0, 1, \ldots, |C| - 1\}$ without any prior training on those specific classes. CLIP addresses this challenge by framing the classification task as a problem of similarity measurement

between textual and visual feature representations, where each class descriptor $\delta_i$ (e.g., "Cat", "Elephant") is represented by a text embedding vector. These vectors are obtained by encoding class descriptors using CLIP's text encoder, typically in the form of a structured prompt such as "This is an image of a $[\delta_i]$". This process produces an embedding vector $w_i \in \mathbb{R}^d$ for each class $i$. These vectors are then normalized to create text-based class reference vectors $\hat{w}_i = w_i/\|w_i\|$ on the unit hypersphere. Similarly, an image $I$ is encoded by CLIP's image encoder to obtain an embedding $x$, which is then normalized to $\hat{x} = x/\|x\|$ on the unit hypersphere. The prediction for image $I$ is the class $i$ with the largest cosine similarity $s_i$, which is given by the dot product $s_i = \langle \hat{w}_i, \hat{x} \rangle$. We denote by $W_{\text{text}} \in \mathbb{R}^{|C| \times d}$ the matrix whose rows are $\hat{w}_i$. The class prediction for an image $I$ is thus given by $\arg\max_i W_{\text{text}}\hat{x}$.

## 3.2 CLIP-ENHANCE

We aim to address two main problems. First, text embedding vectors and image embedding vectors for a given class often have lower than desired cosine similarity. To tackle this issue, we enhance the textual descriptors by applying knowledge distillation to the clustering process, thereby improving accuracy. Second, CLIP's training data exhibits a long-tailed distribution which leads to unequal concentrations of image vectors for different semantic classes on the hypersphere. This causes cosine similarity to perform worse as a discriminator for classes with extreme (high or low) concentrations. Inspired by previous work on integrating SSL and vMF distributions (Govindarajan et al., 2022; Scott et al., 2021; Karpukhin et al., 2024) we propose to address these two problems in CLIP zero-shot classification using a von Mises-Fisher mixture model, which provides a principled and flexible approach to modelling representations of various classes.

### 3.2.1 VON MISES-FISHER CLUSTERING

The von Mises-Fisher distribution is a probability distribution on the $d - 1$ dimensional unit hypersphere in $\mathbb{R}^d$. The probability density function at a vector $\hat{x} \in R^d$ with $\|\hat{x}\| = 1$ is given by

$$P(\hat{x}; \hat{\mu}, \kappa) = \mathcal{C}_d(\kappa)\exp(\kappa\langle\hat{\mu}, \hat{x}\rangle),$$

where $\hat{\mu}$ is the mean direction with $\|\hat{\mu}\| = 1$, $\kappa \geq 0$ is the concentration parameter and $\mathcal{C}_d(\kappa)$ is a normalizing constant. $\mathcal{C}_d(\kappa) = \kappa^{d/2-1}/[(2\pi)^{d/2}I_{d/2-1}(\kappa)]$, where $I_\nu$ denotes the modified Bessel function of the first kind at order $\nu$.

We now consider the clustering problem on the unit hypersphere using a vMF mixture model (Banerjee et al., 2005). The parameters of the model consist of mixture weights $\boldsymbol{\pi} = (\pi_i)_{i \in C}$, class mean directions $\hat{\boldsymbol{\mu}} = (\hat{\mu}_i)_{i \in C}$ and concentrations $\boldsymbol{\kappa} = (\kappa_i)_{i \in C}$ for each class component. For any unit vector $\hat{x}$, the probability that $\hat{x}$ belongs to class $i$ is given by the conditional

$$P(y = i \mid \hat{x}; \hat{\boldsymbol{\mu}}, \boldsymbol{\kappa}) = \frac{\pi_i \mathcal{C}_d(\kappa_i)\exp(\kappa_i\langle\hat{\mu}_i, \hat{x}\rangle)}{\Sigma_{j \in C}\pi_j \mathcal{C}_d(\kappa_j)\exp(\kappa_j\langle\hat{\mu}_j, \hat{x}\rangle)}, \tag{1}$$

assuming every class is equally likely to occur, thus we use equal mixture weights among classes, i.e $\pi_i = 1$ for all $i$, we get

$$P(y = i \mid \hat{x}; \hat{\boldsymbol{\mu}}, \boldsymbol{\kappa}) = \frac{\exp\big(\kappa_i\langle\hat{\mu}_i, \hat{x}\rangle + \log\mathcal{C}_d(\kappa_i)\big)}{\Sigma_{j \in C}\exp\big(\kappa_j\langle\hat{\mu}_j, \hat{x}\rangle + \log\mathcal{C}_d(\kappa_j)\big)}. \tag{2}$$

Computing $\mathcal{C}_d(\kappa)$ exactly is intractable in high dimensions. However, Scott et al. (2021) have shown that we can approximate $\log\mathcal{C}_d(\kappa)$ up to a constant $\eta$ which does not depend on $\kappa$,

$$\log\mathcal{C}_d(\kappa) \approx \mathcal{F}_d(\kappa) + \eta.$$

Here, $\mathcal{F}_d(\kappa)$ is given by the following expression which is differentiable with respect to $\kappa$

$$\begin{aligned}
\mathcal{F}_d(\kappa) = {} & \frac{d-1}{4}\log\left(\frac{d-1}{2} + \sqrt{\left(\frac{d-1}{2}\right)^2 + \kappa^2}\right) - \frac{1}{2}\sqrt{\left(\frac{d-1}{2}\right)^2 + \kappa^2} \\
& + \frac{d-1}{4}\log\left(\frac{d-1}{2} + \sqrt{\left(\frac{d+1}{2}\right)^2 + \kappa^2}\right) - \frac{1}{2}\sqrt{\left(\frac{d+1}{2}\right)^2 + \kappa^2}.
\end{aligned} \tag{3}$$

This allows us to use the following approximation of Eq. 2

$$P(y = i \mid \hat{x}; \hat{\boldsymbol{\mu}}, \boldsymbol{\kappa}) \approx \frac{\exp\big(\kappa_i \langle \hat{\mu}_i, \hat{x} \rangle + \mathcal{F}_d(\kappa_i)\big)}{\Sigma_{j \in C} \exp\big(\kappa_j \langle \hat{\mu}_j, \hat{x} \rangle + \mathcal{F}_d(\kappa_j)\big)}. \tag{4}$$

We note that the vMF mixture can be parameterized by a single matrix $W \in \mathbb{R}^{|C| \times d}$, whose rows $w_i$ encode mean directions and concentrations for each class $i$ as follows: $\hat{\mu}_i = w_i / \|w_i\|$ and $\kappa_i = \|w_i\|$. Conversely, any vMF mixture with parameters $\hat{\boldsymbol{\mu}}$ and $\boldsymbol{\kappa}$ corresponds to a matrix $W \in \mathbb{R}^{|C| \times d}$ whose rows are given by $w_i = \kappa_i \hat{\mu}_i$ for each class $i$. Then, writing $\mathcal{F}_d(W) = (\mathcal{F}_d(\|w_i\|))_{i \in C}$, the probability distribution $P_W(\hat{x})$ over classes for the representation can be expressed compactly in matrix form as

$$P_W(\hat{x}) = \sigma(W\hat{x} + \mathcal{F}_d(W)), \tag{5}$$

where $\sigma$ is the softmax activation function. We remark that if all concentration parameters are equal, then in fact $P_W(\hat{x}) = \sigma(W\hat{x})$ and therefore the class with largest conditional probability is simply given by $\arg\max_{i \in C} W\hat{x}$. Hence the extra capacity of the vMF mixture model really lies in the possibility of selecting or learning different concentrations for each class.

### 3.2.2 KNOWLEDGE DISTILLATION

While normalized CLIP representations, given by rows of $\hat{W}_{\text{text}}$, provide crucial prior information about mean directions of the vMF distribution for each class, they do not represent the optimal vectors for computing image similarity, nor do they seem to encode any useful information about concentrations of each class representations. According to our vMF mixture model interpretation, the matrix $W$ can encode both direction and concentration. To learn the optimal mixture model parameters, we propose to use a knowledge distillation pipeline inspired by DINO (Caron et al., 2021).

The teacher and student vMF mixture models are parametrized by matrices $W_t$ and $W_s$, respectively, as in Eq. 5. For simplicity, we write $P_t$ and $P_s$ for $P_{W_t}$ and $P_{W_s}$, respectively. We initialize all models using the same concentration $\kappa_0$ since we assume no prior in class concentrations, thus, initially $W_s = W_t = \kappa_0 \hat{W}_{\text{text}}$. While we are eventually learning different concentrations for each class, by initially fixing identical concentration for all classes, the student and teacher model's predictions agree with that of zero-shot CLIP (Sec. 3.1) at initialization, as noted after Eq. 5. We use the same value for $\kappa_0$ in all our experiments and this choice is briefly discussed in the implementation details § 4.3.

Given an image $I$, we generate a set of diverse student views $v \in V$, by repeatedly copying and applying different augmentaiton such as random resized cropping to the image $I$. Similarly, we create a single teacher view $I_t$. For each student view $v \in V$, the CLIP vision encoder provides a normalized representation $\hat{x}_v$; likewise, for the teacher, $\hat{x}_t$ for $I_t$. These embedding vectors are then fed to the student and teacher models, which produce outputs $P_s(\hat{x}_v)$ and $P_t(\hat{x}_t)$, respectively.

In our context, both the student and the teacher act on representations obtained from CLIP's vision encoder, whose parameters are frozen. The student model is trained to minimize the Kullback-Leibler divergence between a subset of predictions on its most confident views $V^{\text{conf}} \subseteq V$ and the teacher's prediction on $I_t$. More precisely, given a single image $I \in D$, student parameters $W_s$ are updated using stochastic gradient descent with respect to the following loss function:

$$\ell_{\text{KD}}(I; W_s) = \sum_{v \in V^{\text{conf}}} \text{KL}(P_t(\hat{x}_t) \| P_s(\hat{x}_v)), \tag{6}$$

where $V^{\text{conf}}$ represents the a set of highly confident views with smallest entropy $H(P_s(\hat{x}_v))$, as in TPT (Shu et al., 2022). At the end of each epoch, the teacher parameters $W_t$ are updated using the student parameters $W_s$ through an exponential moving average (EMA) with ratio $\alpha$. While in DINO (Caron et al., 2021) predictions are centered and sharpened to avoid representation collapse, we do not rely on these practices in CLIP-Enhance, possibly because the embeddings from the CLIP text encoder provide a strong initialization.

### 3.2.3 CLIP-ENHANCE SYSTEM OVERVIEW

We briefly describe here all the steps of our zero-shot approach to classification with CLIP as presented in Alg. 1. We assume we have access to class descriptors $\delta_i$ for classes $i \in C$ and we are

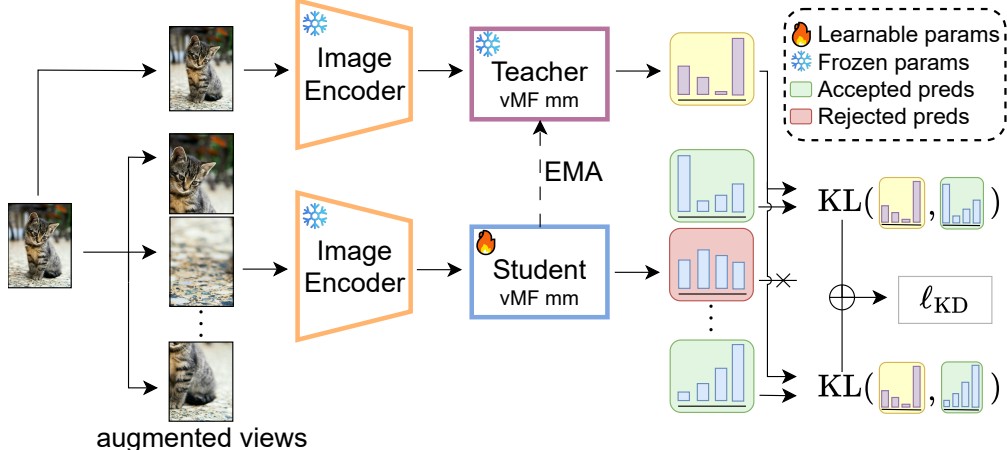

Figure 2: **CLIP-Enhance for zero-shot image classification**. CLIP's frozen image encoder generates embeddings for an input image and its augmented views, which are fed to the teacher and student vMF mixture model, respectively. We compute loss by summing KL divergences between teacher and top-k student predictions, filtered based on entropy. We train only the student model per step, updating the teacher via exponential moving average (EMA) per epoch.

given a (balanced) dataset $D$ of unlabeled images. We first use the CLIP text encoder to compute the matrix $\hat{W}_{\text{text}} \in \mathbb{R}^{|C| \times d}$ whose rows represent our initial class reference directions. This matrix $\hat{W}_{\text{text}}$ serves as an initialization for the student and teacher vMF mixture model $W_s$ and $W_t$, respectively. We then perform our knowledge distillation using the dataset $D$ as presented in § 3.2.2 and illustrated in Fig. 2. Finally, the resulting teacher model is used for inference. Each image is represented as an embedding vector using CLIP, $\hat{x}$, whose class conditional probabilities under the teacher vMF mixture model are given by $P_{W_t}(\hat{x})$ (Eq. 5). The predicted class is the one with maximum probability.

---

**Algorithm 1** CLIP-Enhance

---

1: **Require:** Pre-trained model CLIP, class descriptions $\delta_i$ for each class in $C$, dataset $D$
2: **Output:** List of predicted labels $Y$
3: $W_{\text{text}}[i] \leftarrow \text{CLIP}_{\text{text}}(\text{"This is an image of } [\delta_i]\text{"})$
4: $W_t \leftarrow \text{KD}(W_{\text{text}}, D)$          ▷ Knowledge distillation (§3.2.2 & Alg. 2)
5: **for** $I$ in $D$ **do**          ▷ Inferring over the data set $D$
6:     $\hat{x} \leftarrow \frac{x}{\|x\|}; x \leftarrow \text{CLIP}_{\text{image}}(I)$
7:     $Y[I] \leftarrow \arg\max_{i \in C} P_{W_t}(\hat{x})[i]$          ▷ Prediction from vMF mixture model (Eq. 5)
8: **end for**
9: **return** $Y$

---

## 4 EXPERIMENTS

Our primary hypothesis is that CLIP-Enhance improves classification accuracy in the zero-shot setting compared to state-of-the-art algorithms. To test this, we evaluate performance over a variety of datasets, many of which are commonly used in other zero-shot or few-shot settings (Radford et al., 2021; Gao et al., 2024; Guo et al., 2023; Zhou et al., 2022c;b; Shu et al., 2022). These datasets cover different sets of problems faced in image classification, such as general image classification (ImageNet, CIFAR10, CIFAR100,Caltech101) where the task is to classify images into a large number of classes that are easily differentiable to most humans, as well as datasets designed for fine-grained classification (FGCV-Aircraft, Food101, MNIST, Flowers102,StanfordCars, OxfordIIITPets), where the classifier must distinguish between sub-categories of common objects which are typically hard to differentiate with text (e.g., A-320 vs. A-321 aircraft). Finally, we also test on datasets that may be considered out-of-distribution for CLIP (Image-net-Sketch, EUROSAT, SUN397). We compare against several strong baselines, including TPT (Shu et al., 2022), CALIP (Guo et al., 2023), and CLIP

---

**Algorithm 2** vMF Mixture Model Knowledge Distillation KD($W_{\text{text}}, D$)

---

1: **Require:** Text class reference matrix $W_{\text{text}}$, dataset of images $D$, initial concentration $\kappa_0$,
2: **Require:** Number of epochs $E$, number of views $V$, learning rate $\eta$
3: **Output:** Optimized teacher model parameters $W_t$
4: $W_t, W_s \leftarrow \kappa_0 \hat{W}_{\text{text}}$          ▷ Initialize student, teacher vMF mixture models (§3.2.2)
5: **for** $e$ in $E$ **do**
6:      **for** batch $B$ in $D$ **do**
7:         $L = 0$
8:         **for** $I$ in $B$ **do**
9:            $I_t \leftarrow I, I_v \sim \text{Transform}(I)$ for $v \in V$      ▷ Create views using random transform
10:           $\hat{x}_t \leftarrow \text{CLIP}_{\text{image}}(I_t), \hat{x}_v \leftarrow \text{CLIP}_{\text{image}}(I_v)$ for $v \in V$    ▷ Encode images using CLIP
11:           $P_t \leftarrow P_{W_t}(\hat{x}), P_{s,v} \leftarrow P_{W_s}(\hat{x}_v)$ for $v \in V$      ▷ Compute predictions using Eq. 5
12:           $H_{s,v} \leftarrow - \sum_{i \in C} P_{s,v} \log P_{s,v}$ ▷ Compute and sort entropy of student's predictions
13:           $V^{\text{conf}} \leftarrow$ top 10% of views $v$ with smallest $H_{s,v}$      ▷ Select most confident views
14:           $L \leftarrow \sum_{v \in V^{\text{conf}}} \text{KL}(P_t \| P_{s,v})$         ▷ Accumulate loss over the batch
15:           $W_s \leftarrow W_s - \eta \nabla_{W_s} L$         ▷ Gradient update of student parameters
16:         **end for**
17:         $W_t \leftarrow \alpha W_t + (1 - \alpha) W_s$       ▷ Update teacher using exponential moving average
18:      **end for**
19: **end for**
20: **return** $W_t$

---

zero-shot (Radford et al., 2021). These approaches, similar to CLIP-Enhance, are truly zero-shot in that they require no external information apart from the textual description of classes to classify the images.

## 4.1 DATASETS

We evaluate CLIP-Enhance on a total of 15 datasets: CIFAR10 (Krizhevsky et al., 2009), CIFAR100 (Krizhevsky et al., 2009), ImageNet (Deng et al., 2009a), DTD (Cimpoi et al., 2014), EuroSAT (Helber et al., 2018; 2019), Food101 (Bossard et al., 2014), Flower102 (Nilsback & Zisserman, 2008), FGVC-Aircraft (Maji et al., 2013), SUN397 (Xiao et al., 2010), MNIST (LeCun et al., 2010). UCF101 (Kay et al., 2017), Caltech101 (Li et al., 2022a) OxfordPetsIIIT (Parkhi et al., 2012), Stanford-Cars (**?**) Imagenet-sketch (Wang et al., 2019) We follow the standard zero-shot evaluation protocols outlined in the original CLIP paper (Radford et al., 2021).

## 4.2 ZERO-SHOT BASELINES

We compare CLIP-Enhance against other state-of-the-art zero-shot methods, CALIP (Guo et al., 2023) and TPT (Shu et al., 2022), as well as CLIP zero-shot (Radford et al., 2021) (§3.1). These methods require no additional information, apart from text labels for classes. CALIP (Guo et al., 2023) proposes an image-text integrated attention module in CLIP's transformer backbone, such that both image and text attention masks are functions of both image and text features. TPT (Shu et al., 2022) proposes, for each sample $I \in D$, to learn an adaptive prompt at test time. First, a mini batch of embeddings $\{x_1, x_2 ... x_n\}$ is obtained by encoding $n$ randomly augmented views of image $I$, and the average entropy across the views whose predictions are most confident is minimized via single step of gradient descent. Predictions are then made using the adapted prompt by averaging the predictions from high-confidence views.even though TPT works under the Test time paradigm i.e it evaluates and optimizes its prediction one image it still falls under the zero-shot classification.

## 4.3 IMPLEMENTATION DETAILS

We use CLIP's available base model[1], and use an RN50 backbone for the vision encoder, due to its light weight, ease of implementation, and widespread use among other state-of-the-art systems. For CLIP-Enhance, following Radford et al. (2021), we resize all test images to $224 \times 224$ resolution

---

[1]https://github.com/openai/CLIP

Table 1: **Classification Accuracy in %** (↑) of various CLIP zero-shot image classification methods. We report the results for the baselines (CLIP ZS (Radford et al., 2021), CALIP (Guo et al., 2023), TPT (Shu et al., 2022)) from the original papers. The best method for a given dataset is in bold.

| Method | C-10 | C-100 | MNIST | Fl-102 | SUN397 | DTD | EUROSAT | FGVCA | Food101 | UCF101 | Ctech | Pets | S-Cars | ImageNet | I-Sketch |
|---|---|---|---|---|---|---|---|---|---|---|---|---|---|---|---|
| CLIP ZS | 71.3 | 40.8 | 55.8 | 65.8 | 58.5 | 41.5 | 37.5 | 17.1 | 77.3 | 58.84 | 85.88 | 83.57 | 55.70 | 60.3 | 33.37 |
| CALIP | N/A | N/A | N/A | 66.4 | 58.6 | 42.4 | 38.9 | 17.8 | 77.4 | 61.72 | **87.71** | **86.21** | 56.27 | 60.6 | 35.6 |
| TPT | N/A | N/A | N/A | 62.7 | **61.4** | 40.8 | 28.7 | 17.6 | 74.9 | 60.82 | 87.02 | 84.49 | **58.46** | 60.7 | 35.09 |
| CLIP-Enhance | **80.3** | **46.4** | **65.0** | **67.3** | 59.6 | **43.3** | **39.4** | **18.3** | **81.6** | **65.71** | 86.4 | 85.53 | 56.7 | **61.4** | **36.1** |

and use a batch size of $512$. We run CLIP-Enhance for only 20 epochs, and use $\alpha = 0.99$ in the EMA weight update for all experiments across all datasets. We use $64$ views and for entropy filtering we select the top $10\%$ most confident views (views with the lowest entropy). We found that using the identity augmentation for the teacher model and using only random-resized-crop transformation with scale ranging from $0.6$ to $0.8$ for the student model lead to the best performance. In all our experiments, we run CLIP-Enhance (Alg. 1) on the test set.

For the initial concentration value, we use the same value $\kappa_0 = 3000$ for all experiments. Recall from Eq. 4 that each logit takes the form $\kappa c + \mathcal{F}_d(\kappa)$ where $c \in [-1, 1]$ is the cosine similarity. Empirically we observe that cosine similarities between text embeddings and image embeddings with CLIP typically vary in the range $[0.2, 0.4]$, which corresponds to angles in degrees in $[66°, 78°]$. Our choice $\kappa_0$ is the unique value that ensure that $\kappa_0 \cos(68°) \approx \mathcal{F}_{1024}(\kappa)$, ensuring that both terms in the logit have roughly the same magnitude. Moreover, we empirically find that direct estimation of concentration of representations of images for various classes in CIFAR-10 range between from $4000$ to $6000$, which makes our choice of $\kappa_0$ look like a low concentration in comparison. In addition, we include results for other values of $\kappa_0$ in Appendix A.2.

## 4.4 PERFORMANCE ANALYSIS

Table 1 presents our main results. We observe that CLIP-Enhance significantly enhances zero-shot classification performance compared to CLIP zero-shot for all data sets, particularly on CIFAR10, MNIST, UCF101 and Food101. We hypothesize that this is because the text descriptions of the classes in these datasets are better differentiated in embedding space, possibly due to being more prevalent within CLIP's training data. We can clearly see that CLIP-Enhance outperforms CLIP zero-shot on all datasets, including Imagenet. We also compare against CALIP (Guo et al., 2023), where again we find our approach performs better in all except two Caltech and OxfordIIITpets where CALIP outperforms both our CLIP-Enhanceand TPT. However, we note that CALIP, which proposes fusing multi-modal attention modules for inference, is actually complementary to our method and that both methods could, in theory, be combined. Furthermore, our knowledge distillation approach also improves over the online test-time adaptation setup of TPT (Shu et al., 2022) though TPT outperorms us on the texture classification dataset, SUN397, Caltech101, and Stanford Cars. We hypothesize part of our advantage over TPT is due to our ability to operate directly in embedding space rather than prompt space, which may give CLIP-Enhance more fine-grain control over model parameters. Overall, the consistent, and at times significant, enhancement of zero-shot classification performance across a large number of varied datasets establishes the effectiveness and generalizability of CLIP-Enhance.

## 5 ADDITIONAL ANALYSIS AND ABLATION STUDIES

To provide additional support for modeling the zero-shot clustering problem as a vMF mixture model CLIP-Enhance as well as some of our design choices, we conduct several additional experiments (§5.1) and ablation studies (§5.2,5.3) on a benchmark of 7 data sets including Flowers102 (Nilsback & Zisserman, 2008), DTD (Cimpoi et al., 2014), EuroSAT (Helber et al., 2019), CIFAR10 (Krizhevsky et al., 2009), CIFAR100 (Krizhevsky et al., 2009), MNIST (LeCun et al., 2010), FGVC-Aircraft (Maji et al., 2013), and Food101 (Bossard et al., 2014). Overall, we demonstrate the importance of our vMF mixture model formulation in improving the performance. Furthermore, we empirically show that CLIP's image-text embeddings are mis-aligned and there is a re-alignment of vectors occurring during training that leads to better performance. Additionally, our experiments establish that CLIP-Enhance is remarkably robust to data availability.

Table 2: **Alignment in degrees** (↓) between text and image embeddings, averaged across all classes of a dataset, before and after training with CLIP-Enhance.

| Method | C-10 | C-100 | MNIST | F-102 | DTD | FOOD101 | FGVC |
|---|---|---|---|---|---|---|---|
| CLIP ZS | $78.9°$ | $78.9°$ | $77.4°$ | $73.1°$ | $74.2°$ | $72.5°$ | $\mathbf{73.8°}$ |
| CLIP-Enhance | $\mathbf{78.5°}$ | $\mathbf{78.7°}$ | $\mathbf{76.1°}$ | $\mathbf{72.9°}$ | $\mathbf{74.1°}$ | $\mathbf{72.1°}$ | $73.8°$ |

## 5.1 INCREASED ALIGNMENT THROUGH KNOWLEDGE DISTILLATION

In this section, we make a first step towards providing empirical evidence for our claim from the introduction about the anisotropic character of the distribution of image embeddings around the average text embedding for a given class. Specifically, we compute a metric we call 'alignment', which measures how image representations for each class are aligned with the corresponding reference vector. More precisely, given a matrix $W \in \mathbb{R}^{C \times d}$ and a class $i$ in a dataset $D$, we compute the average cosine similarity between the row $w_i$ – which is our reference direction for class $i$ – and each normalized representations of images in that class. By linearity of the dot product, this is equal to the cosine between the mean direction $\mu_i$ of image representations for class $i$ with the row $w_i$ of $W$, whose corresponding angle we denote by $\alpha_i$. Finally the alignment of $W$ on $D$ is obtained by averaging angles $\alpha_i$ over classes $i$.

In Table 2, we report for each dataset the alignment for CLIP Zero-Shot using $\hat{W}_{\text{text}}$ and for CLIP-Enhance using the matrix $W_t$ learned from $\hat{W}_{\text{text}}$ using Algo 1. We observe that while the alignment may seem low in all cases given the high dimensionality ($d = 1024$) of representations, our knowledge distillation method consistently improves the initial alignment of CLIP zero-shot given by embbedings of text descriptions. Although, the absolute difference in degree is relatively small, this seems significant given the high dimension of the representations. This therefore seems to indicate that on average the image representations of each class are distributed closer to the directions learn by CLIP-Enhance than to initial representations obtained from class descriptions. This increased alignment explains partly the performance improvement obtained by CLIP-Enhance.

## 5.2 LEARNING CLASS CONCENTRATIONS MATTERS

In this section we aim to showcase the importance of learning concentration parameters $\kappa_i$ for each class along with the reference directions $w_i$. To do so, we modify our CLIP-Enhance approach by constraining each row to norm 1 and compare the results with our approach. Concretely, we modify Algo 2 by initializing $W_s$ and $W_t$ simply to $\hat{W}_{\text{text}}$, we normalize rows of $W_s$ after each gradient update and proceed similarly after each update of $W_t$ through exponential moving average. As noted after Eq. 6, this normalization process which leads to constant concentrations across classes somehow removes the vMF mixture model component of our approach. Results are presented in Table 3 where performance of this constrained version of our approach are showcased in the "CLIP-Enhance w/o vMF" row.

While this constrained version of our approach does improve slightly on the initial CLIP zero-shot, when comparing with CLIP-Enhance's performance we concur that learning concentrations matters a lot and that concentration parameters are crucial in our approach. The advantage of the extra capacity of the vMF mixture model is particularly clear on CIFAR10, CIFAR100 and MNIST. Further, in table 3 we also showcases the performance gain achieved due to multiview ensebling evaluation. we observe that XXX

## 5.3 AFFECT OF LIMITED DATA

Here in table 4, we explore the influence of data availability on CLIP-Enhance's performance, as data set size is known to play a significant role in the success of SSL approaches (Kaplan et al., 2020). We keep all the parameters and implementation details the same as in earlier experiments (Alg. 1 and §4.3) except we only use $100\%$, $50\%$, $10\%$, and $0\%$ (representing CLIP zero-shot) of the data for knowledge distillation (Alg. 1, Line 4). We observe a drop in performance when using $50\%$ of the test data as compared to $100\%$ test data availability, though not all data sets are affected equally. For

Table 3: Ablation study of CLIP-Enhance. "CLIP ZS" refers to the CLIP model without any zero-shot training, "CLIP-Enhance w/o vMF" refers to the CLIP model with self-supervised training without using vMF mixture model, and "CLIP-Enhance" refers to the proposed variant of CLIP with self-supervised training using vMF mixture model.

| CLIP Variant | C-10 | C-100 | MNIST | F-102 | DTD | FOOD101 | FGVC |
|---|---|---|---|---|---|---|---|
| CLIP ZS | 71.3 | 40.8 | 55.8 | 65.8 | 41.5 | 77.3 | 17.1 |
| CLIP-Enhance w/o vMF | 74.5 | 42.7 | 57.7 | 67.0 | 41.5 | 77.8 | 17.6 |
| CLIP-Enhance w/o Ensembling | 79.6 | 45.8 | **65.1** | 67.3 | 42.8 | 80.2 | 17.9 |
| CLIP-Enhance | **80.3** | **46.4** | 65.0 | **67.3** | **43.3** | **81.6** | **18.3** |

Table 4: Effect of varying data availability on CLIP-Enhance.

| Data avail. | C-10 | C-100 | MNIST | F-102 | DTD | FOOD101 | FGCV |
|---|---|---|---|---|---|---|---|
| 0% | 71.3 | 40.8 | 55.8 | 65.8 | 41.5 | 77.3 | 17.1 |
| 10% | 78.2 | 41.5 | 57.8 | 66.8 | 41.5 | 79.8 | 17.2 |
| 50% | 78.9 | 42.6 | 57.9 | 66.9 | 41.6 | 80.5 | 17.3 |
| 100% | 80.3 | 46.4 | 64.9 | 67.3 | 43.3 | 81.6 | 18.3 |

example, MNIST and CIFAR100 exhibit significant decreases in accuracy while most other data sets remain much less affected. Overall, we see that CLIP-Enhance can still improve performance given even a small fraction of the data, and in some cases, such as C-10 and FOOD101, improvements can be significant.

# 6 CONCLUSION AND LIMITATIONS

This paper proposes a novel method, CLIP-Enhance, for improving CLIP zero-shot classification, by reformulating it as von Mises-Fisher mixture model. CLIP-Enhance leverages CLIP's invariant representation space to optimize this von Mises-Fisher mixture model in order to mitigate inherent misalignment between embedding vectors of different data modalities as well as model the non-uniformity of embedding concentrations between different classes. We show the relative efficacy of CLIP-Enhance compared to standard CLIP zero-shot classification as well as other state-of-the-art methods on a number of standard datasets. We also show that CLIP-Enhance provides performance gains even under constrained conditions, where only a small fraction of the test set is available or training is conducted over a small number of epochs.

While providing a significant improvement over CLIP zero-shot, CLIP-Enhance does have some limitations. Perhaps the most obvious, and the most widely shared limitation amongst all CLIP fine-tuning, prompt engineering, adapter, or post-processing methods is that we are limited by the quality of CLIP's original representation space. First, the relatively poor performance of our initialization using representations of class descriptions may impede our knowledge distillation step. Second, the quality of CLIP image representations may limit the performance that methods like CLIP-Enhance can achieve. Additionally, although we show improvements even with small amounts of data, CLIP-Enhance still does significantly better when given access to more data, and it is likely that this tradeoff could be further improved. We see a similar trend with epochs trained, where longer training runs provide better performance. Possibly, performing knowledge distillation with a small amount of data on which the initial vMF classifier shows better accuracy would lead to interesting results, but this would require an efficient proxy for the uncertainty of multi-modal models like CLIP. Overall, zero-shot classification in this context remains a very challenging task.

In future work we plan to investigate extensions to few-shot learning as well as exploring a greater number of SSL algorithms which may further close the gap between zero-shot CLIP-Enhance and linear probe classifiers trained on the full dataset. It may also be possible to use the output of algorithms like CLIP-Enhance to evaluate or improve the CLIP's representation space, creating a closed-loop system, though this possibility has yet to be seriously explored.

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

# A  APPENDIX

## A.1  SUPERVISED BASELINES:

In 5 we evaluate the performance of supervised baselines including image-center clustering and a linear probe, These supervised baselines establish classification upper bounds for the particular dataset, indicate the extent to which the data is easily separable, and generally give some insight into the difficulty of the dataset. We establish an upper bound for classification performance using a linear probe, which are widely regarded as a strong baseline for assessing the potential of pre-trained representations for a specific task, especially in scenarios with limited data available for training the classifier (e.g., zero-shot learning) (Tian et al., 2020).

Next, we perform classification using image mean clustering, where cluster directions are computed by averaging the image embeddings belonging to each class within the dataset. These image cluster directions are then used to classify the test set. the aim of this is to establish the magnitude of alignment error, which refers to the discrepancy in image embeddings and textual embeddings describing the same image, do not point in the same direction. this also co-relates with lower alignment within text embedding vector and image embedding vectors. for example in MNIST where its hard to differentiate the image of digits through a textual description on a image of digit ( image of number five vs image of number four) , we see a huge performance gain when using Image mean cluster center but in Datasets such as Food101 , CIFAR100 where the classes are quite different even in the text domain( apple pie vs hot dog, CAT vs DOG). Finally, we also explore non-linear decision boundaries, by studying the performance of K nearest neighbours, where classification is done by finding the most common class among the K nearest neighbours( Top K neighbours with the highest cosine similarity).

Table 5: Performance of CLIP-Enhance compared to vanilla CLIP zero-shot and supervised baselines.

| Group | Method | C-10 | C-100 | MNIST | F-102 | DTD | FOOD101 | FGCV |
|---|---|---|---|---|---|---|---|---|
| CLIP | zero-shot | 71.3 | 40.8 | 55.8 | 65.8 | 41.5 | 77.3 | 17.1 |
| Oracles | KNN | 82.3 | 55.7 | 96 | 71.3 | 61.7 | 79.5 | 29.8 |
| | Image Center | 77.1 | 51.1 | 78.4 | **90.8** | 61.5 | 78.7 | 32.3 |
| | Linear Probe | **86.7** | **63.6** | **95.6** | 85.6 | **65.8** | **85.1** | **33.8** |
| Ours | CLIP-Enhance | 80.3 | 46.4 | 65.0 | 67.3 | 43.3 | 81.6 | 18.3 |

## A.2  DIFFERENT VALUES FOR INITIAL CONCENTRATION $\kappa_0$

In this section, we provide some results on the impact of different choices for the initial concentration parameter $\kappa_0$. Results from $\kappa_0 = 500$ and $\kappa_0 = 5000$ are provided in Table 6 along with the results of the value we used everywhere else in the paper, namely $\kappa_0 = 3000$. Building on the discussion in § 4.3 regarding the choice of $\kappa_0$, we further illustrate in Figure 3 how increasing $\kappa_0$ reduces the contribution of the normalization term to the logit score. This observation provides insight into the observed performance degradation: as the model increasingly relies on the normalization component rather than the image data for prediction its performance declines, around the $\kappa_0 = 2000$ the contribution reaches close to zero and we observe that model accuracy drop near the values obtained by our model without Vmf formulation as discusess in section § 5.2 . Additionally, we observed that larger values of $\kappa_0$ result in numerical instability during training in our setup, leading to a significant drop in performance.

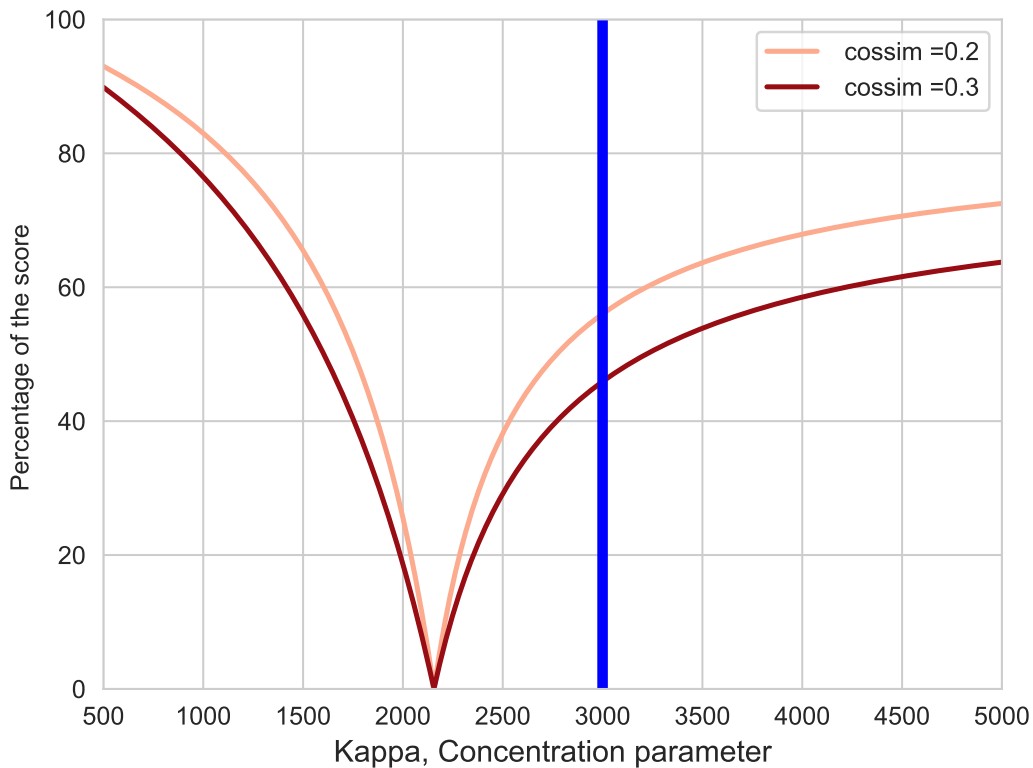

Figure 3: Contribution by the Normalization term as seen in equation 5 towards the whole logit term, As the $\mathcal{F}_d(W)$ decrease with the increase of Kappa we first observe a decrease in contribuiton which later flip as the normalization term starts becoming negative.

Table 6: Effect of varying $\kappa_0$ on CLIP-Enhance.

| $\kappa_0$ | C-10 | C-100 | MNIST | F-102 | FOOD101 |
|---|---|---|---|---|---|
| 1000 | 78.7 | 43.5 | 56 | 65.8 | 78.4 |
| 2000 | 75.6 | 43.1 | 57.7 | 67.1 | 78.1 |
| 3000 | **80.3** | **46.4** | **64.95** | **67.3** | **81.6** |
| 4000 | 79.7 | 46.2 | 64.8 | 66.3 | 78.4 |
| 5000 | 79.8 | 46.1 | 53.1 | 11.3 | 76 |

## A.3 HOW CLIP-ENHANCEIMPROVES PERFORMANCE

In Figure 4, we illustrate the performance of different prediction models:(1) CLIP's native zero-shot classifier, (2) a simple SSL linear classifier trained using our SSL approach, and (3) our proposed CLIP-Enhancemodel.

Examining the confusion matrix for CLIP ZS, we observe a tendency to over predict certain classes, such as cars and birds, while significantly under predicting others, including frogs, trucks, and deer. We hypothesize that this behavior arises from data imbalance in CLIP's training dataset, where certain classes are overrepresented, causing the model to favor them even during inference.

In contrast, the linear classifier model , which is learns a simple unconstrained classification matrix W through our SSL process, improves the accuracy considerably , thus showcasing the our SSL's ability to align the modalities. Finally, our CLIP-Enhance model shows a more uniform prediction accuracy across classes. Previously underrepresented classes, such as frogs , are predicted more frequently, while overrepresented classes, like cars, are predicted less often. This suggests that CLIP-Enhance

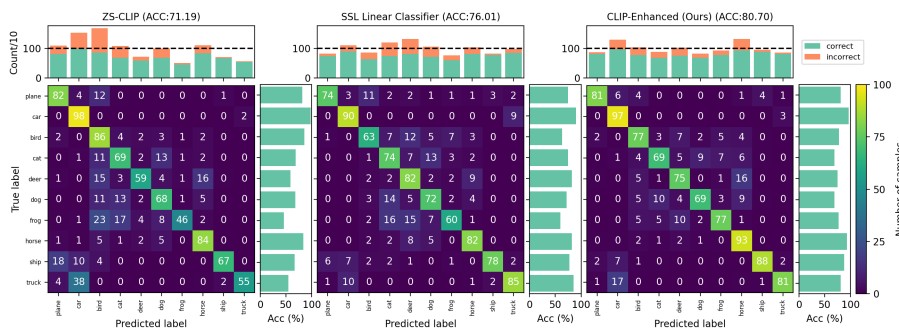

Figure 4: The figure illustrates the evolution of the confusion matrix on the CIFAR10 dataset, comparing CLIP's zero-shot classification with our CLIP-Enhanceapproach. The top bar plot displays the total class-wise predictions made by each model, while the right-side bar plot depicts the class-wise accuracy. We evaluate three models: (1) CLIP's native zero-shot classifier, (2) a simple SSL linear classifier trained using our SSL approach, and (3) our proposed CLIP-Enhancemodel.

effectively mitigates class imbalance, by learning a lower concentration Kappa (ref table 7for these underrepresented classes such as deer and frog.

| CIFAR-10 Class | Learned Concentration |
|---|---|
| Plane | 3032.22 |
| Car | 3046.86 |
| Bird | 3070.32 |
| Cat | 3058.59 |
| Deer | 2964.84 |
| Dog | 3073.23 |
| Frog | 2951.67 |
| Horse | 3032.22 |
| Ship | 3032.22 |
| Truck | 3023.43 |

Table 7: Learned concentration values for CIFAR-10 classes by our CLIP-Enhance

### A.3.1 REDEFINING ALIGNMENT

CLIP's training objective aims at a relative alignment of captions embeddings with embeddings of their corresponding image embeddings (Radford et al., 2021). More precisely, for a random batch from CLIP's training dataset, the symmetric cross entropy is minimized when 1): every caption embedding is more are aligned (larger cosine similarity) with its corresponding image embedding than with any other image embedding 2) every image embedding is more are aligned (larger cosine similarity) with its corresponding caption embedding (positive pair) than with any other caption embedding (negative pairs). Moreover, since "a temperature parameter which controls the range of the logits in the softmax is directly optimized during training" – which would appear to take approximately the value $\exp(\tau) = 100$ [1], which is the maximal value allowed during training to prevent instability (Radford et al., 2021) – a small difference between the cosine similarity of a positive pair compared to negative pairs leads to cross entropy loss which is roughly zero. For instance, for the training batch size ($b = 32,768$), a positive cosine similarity of $0.5 = \cos(60°)$ for negative cosine similarities of $0.225 \approx \cos(77°)$ obtains a cross entropy loss (when logits are scaled by 100) which is smaller than the smallest representable number (with Pytorch, Float32).

Hence, CLIP's training does not aim at aligning text embeddings and their corresponding images in the sense that the angle between them is small (cosine almost 1). On the contrary, in CLIP's multimodal embedding space, the angle between a text representation and its corresponding image representation can be as large as $60°$ (cosine $0.5$) as long as negative pairs are less aligned, namely with an angle

larger than $77°$. Therefore, we conclude that the quality of CLIP's multimodal embeddings really resides in their RELATIVE alignment, not in the actual alignment of positive pairs, an observation that is also supported by (Liang et al., 2022).

Furthermore, we consider that this relative alignment highly depends on the training distribution. When leveraging these representations for a downstream classification task, we aim at a relative alignment of text-based class reference embeddings with respect to the image distribution. A specific balanced classification task like CIFAR10 is expected to differ greatly from the long tail distribution of CLIP's training dataset, resulting in a poor relative alignment. Our approach aims at adapting in an unsupervised manner the initial text-based class reference embeddings to improve their relative alignment on the image dataset. By endowing each class reference with a concentration parameter and formulating the classification problem as a von Mises–Fisher mixture model, our approach achieves this goal as reflected by the improved accuracy. [2]

---

[2][1] CLIP's official git repository. API. url:https://github.com/openai/CLIP.

