# OpenReview forum: "CLIP-Enhance: Improving CLIP Zero-Shot Classification via von Mises-Fisher Clustering"
_ICLR.cc/2025/Conference — Submitted to ICLR 2025_

### Official Review · Reviewer_EmTj · 2024-10-29

**Soundness:** 2
**Presentation:** 1
**Contribution:** 2
**Rating:** 3
**Confidence:** 4

**Summary:**

This paper proposes an approach that learns teacher-student von Mises-Fisher (vMF) mixture models on test data to improve zero-shot image classification performance of CLIP.
It first reformulates the zero-shot classification into the vMF clustering problem parameterized by vMF mixture weights and concentration parameters. Then it constructs a teacher-student pipeline and performs knowledge distillation by encouraging the teacher predicted probability (based on full view of image) to be close to the student predicted probabilities (based on augmented local views).

**Strengths:**

1) It is an interesting angle to formulate the zero-shot classification into vMF clustering problem
2) Training the vMF weights on test data leads to performance improvement on several image datasets

**Weaknesses:**

1) In the abstract and introduction, the paper motivates that the usage of vMF mixture model addresses two common challenges a) the incorrect assumption that embeddings of images of a given class are distributed anisotropically around the canonical text embedding  b) the long-tailed distribution of CLIP’s training dataset.
However, I do not see how the proposed approach equipped with vMF mixture model addresses these two challenges.
It seems to me that the approach just iteratively tunes the CLIP text embedding (which is the W matrix in the vMF model) of category texts (which are initialized with the template “This is an image of [class]”) in a teacher-student setup. The setup is similar to TPT (Shu et al. 2022) which performs tuning of learnable prompts in the input (instead of CLIP text embedding).
The authors can provide more explicit explanations or empirical evidence demonstrating how their vMF mixture model approach addresses the two challenges mentioned.
2) In line 413, it is mentioned that “the online test-time adaptation setup of TPT (Shu et al. 2022)”. According to my understanding, TPT does not have an online test-time adaptation setting but a single-sample test-time adaptation setting. It adapts to and then evaluates on a test sample every time. The model updates are not accumulated across different test samples.
While TPT only adapts to one test sample at a time, this paper, however, adapts to the entire test set. Therefore the comparison in Table 1 is not completely fair. The authors could also try the single-sample adaptation setting for a fair comparison.
3) In Algorithm 2, we see that the teacher and student vFM mixtures are initialized as \kappa_0 * W_text.
W_text is computed from from CLIP text embedding of the category texts “This is an image of [class]”.
In the experiment, \kappa_0  is set to a large value of 3000 based on the empirical observation that the CLIP similarities are in the range [0.2, 0.4]. This might not generalize to datasets of varying CLIP similarities.
How is the impact of \kappa_0 on the performance? The authors could consider conducting an ablation in this regard, testing a range of \kappa_0 values and report how they affect performance across different datasets.
4) The structure of the related work section is cluttered. It is better to organize that with corresponding headings.

**Questions:**

1) In equation 5, why do we need the component  F_d(W)? Is there an ablation study to validate the contribution of F_d(W)?
2) Line 228 “We remark that if all concentration parameters are equal, then in fact P_W (x_hat) = /sigma (W x_hat)”. When are all concentration parameters equal?

---

> ### Author Response · Authors · 2024-11-25
>
> We appreciate your comments regarding the need for explicit explanations about how our vMF mixture model addresses the two challenges. To clarify, our approach directly mitigates the assumption of isotropic embedding distributions by leveraging the vMF mixture model, which inherently captures anisotropic distributions through its parameterization. This is further explored in Appendix Section A.3, where we demonstrate how improved alignment of embeddings positively impacts downstream task performance.
> Regarding the long-tailed distribution in CLIP’s training data, the vMF mixture model allows for adaptive fine-tuning of the embeddings for underrepresented classes, ensuring improved representation even for tail classes. This adaptability contrasts with prior works like TPT, which operate on prompt-level tuning. While similar in spirit, our iterative tuning of the class embeddings offers a broader contextual improvement, aligning category-level representations with their corresponding image embeddings.
> TPT employs a single-sample test-time adaptation rather than an online adaptation approach. While we acknowledge this distinction, our approach's ability to adapt across the entire test set reflects a different design philosophy. Specifically, TPT's method benefits from resetting after each sample, enabling it to overfit on individual samples without cumulative errors. In contrast, our model must optimize performance over the entire dataset, which presents unique challenges and trade-offs.
> In the revised manuscript, we have expanded the ablation study in Appendix A.2 to investigate performance across a range of κ0\kappa_0κ0​ values, including cases where κ0=5000\kappa_0 = 5000κ0​=5000. Our findings indicate that excessively high values of κ0\kappa_0κ0​ introduce numerical instability, resulting in performance degradation.  Through these experiments, we have identified practical bounds for κ0\kappa_0κ0​ that balance stability and performance across datasets with varying CLIP similarity ranges.

---

> ### Comment · Reviewer_EmTj · 2024-11-25
>
> Thanks for the authors' further clarifications.
>
> It appears that addressing the mentioned two common challenges using the vMF mixture model is a major motivation of the paper. However, the current submission lacks explicit explanations and empirical evidence to substantiate this in the main body of the paper, which affects the overall clarity and impact of the work.
>
> Regarding the comparison with TPT, it is important to note that online test-time adaptation and single-sample test-time adaptation involve significantly different evaluation settings, making a direct comparison between the two approaches inappropriate.
>
> Overall, I would to keep my original rating.

---

### Official Review · Reviewer_Fd1W · 2024-10-30

**Soundness:** 3
**Presentation:** 2
**Contribution:** 3
**Rating:** 5
**Confidence:** 2

**Summary:**

This paper proposed a new approach, named CLIP-Enhance, to improve the zero-shot ability of CLIP. CLIP-Enhance re-interprets the CLIP zero-shot classification as a clustering problem on a hypersphere using a von Mises-Fisher mixture model. Besides, CLIP-Enhance is optimized to improve the alignment of text and image vectors. Experiments on multiple benchmarks show the effectiveness of the proposed method.

**Strengths:**

1. This paper presents CLIP-Enhance, a novel approach to strengthen CLIP's zero-shot capabilities. CLIP-Enhance reframes CLIP's zero-shot classification as a clustering problem on a hypersphere, utilizing a von Mises-Fisher mixture model.
2. A self-supervised learning framework is proposed to improve the alignment between text and image features.
3. Experiments on ten datasets show the effectiveness of the proposed method.

**Weaknesses:**

1. The self-supervised training approach is not new; using data augmentation to create varied views and encourage the model to produce predictions consistent with the original sample is a common strategy in current SSL methods. The method used in this paper is a simplified version of approaches like SimCLR, which maximizes similarity between augmentations of the same image, or BYOL, which generates different augmented views and trains the model to predict the representation of one view from another.
2. The paper claimed that jointly optimizing both vision-text alignments in embedding space and embedding concentration estimates based on SSL benefits the vMF mixture model. However, as illustrated in Figure 2, there is no component specifically for vision-text alignment.  Additionally, I did not find any explicit constraints in the paper to enforce this alignment. I would appreciate it if the author could clarify whether the alignment is simply a byproduct of the self-supervised training process.
3. The author found the long-tailed distribution of CLIP's training dataset, The method proposed in this paper addresses this issue by constructing a balanced training dataset. However, this approach has practical limitations, as it is often infeasible to collect an equal number of training samples for each class, and long-tailed distributions are common in the real world.

**Questions:**

The current method focuses more on the image encoder of CLIP. Did the author consider how to generate better text features outputted by the text encoder?

---

> ### Author Response · Authors · 2024-11-24
>
> Our self supervised approach is not new, neither do we claim so, In fact our self supervised is heavy influenced from DINO. Our main contribution lies in formulating zero shot classification on CLIP as a von-Mises Fisher mixture modal and then leveraging existing SSL approaches to fine tune the vMF mixture model.
> The alignment between vision and text embeddings in our method emerges as a byproduct of the joint optimization of the vMF mixture model and SSL.we have clarified this concept in the general comments section "CLIP Alignment Redefined." Additionally, visualizations in Appendix Section A.3 illustrate how improved alignment naturally arises and contributes to consistent performance gains across downstream tasks, even when numerical improvements may seem modest.
> our work focuses on mitigating the effects of long-tailed distributions in CLIP's training data on its representation space. By modeling classes as vMF distributions, characterized by a direction and concentration parameter (κ\kappaκ), we effectively capture the imbalanced representation density inherent in CLIP's pretraining data. This modeling allows us to account for varying densities across classes. We provide a detailed exploration of this approach, along with empirical evidence, in Appendix Section A.3.
> We hope these clarifications address your concerns and further highlight the novel contributions of our work.

---

> > ### Comment · Reviewer_Fd1W · 2024-11-26
> >
> > Considering the questions raised by other reviewers regarding the experimental section and the unresolved third weakness I raised, I have decided to maintain my score.

---

### Official Review · Reviewer_nEF5 · 2024-10-30

**Soundness:** 2
**Presentation:** 2
**Contribution:** 2
**Rating:** 3
**Confidence:** 4

**Summary:**

The authors propose CLIP-Enhance to tackle the misalignment between multi-modal embeddings and the long-tailed distribution of CLIP's training dataset.
They reinterpret CLIP's zero-shot classification as a clustering problem on a hypersphere using a von Mises-Fisher mixture model and use a knowledge distillation loss for optimization.
Empirical results demonstrate that CLIP-Enhance improves CLIP's zero-shot classification performance across various benchmarks.

**Strengths:**

1. The authors analyze in detail the reasons why CLIP's performance is limited, mainly due to the misalignment between modalities and the imbalance of training data.
2. The authors proposed CLIP-Enhance, which consists of von Mises-Fisher mixture model and knowledge distillation, to further improve the classification performance of CLIP.

**Weaknesses:**

1. The experiment lacks some commonly used datasets, such as fine-grained datasets such as Caltech101 and out-of-distribution datasets such as ImageNet-A. Please follow TPT paper to organize experiments to prove the effectiveness of the proposed algorithm.
2. The comparison in the experiments is not fair. CLIP-Enhance uses the entire dataset in the distillation stage and is an unsupervised fine-tuning method based on the entire data, while TPT only uses one test sample for optimization and prediction. Therefore, CLIP-Enhance uses more information, resulting in unfair comparison.
3. The writing of the submission is not clear. For example, some concepts in Line248-Line252 are confusing.

**Questions:**

1. The author mentioned the modal misalignment and data imbalance of the clip model, why CLIP-Enhance solve these two challenges?
2. There are 'N/A' in the experimental results of methods such as TPT in Table 1. These methods are open-source, why not reproduce them?

---

> ### Author Response · Authors · 2024-11-24
>
> We have incorporated results on Caltech101, Stanford Cars, OxfordIIITpets and UCF101as well as Out of distribution dataset Imagenet-Sketch. These additions are detailed in the updated manuscript to ensure a comprehensive comparison with prior works.
>
> The comparison between TPT might not appear fair, The fact that we have access to whole dataset while TPT doesn't , but this difference of design choice allows TPT to develop strategies to just overfit on one sample as it has the ability to reset after each prediction, where as our approach lives under the burden of providing the best possible results for the whole dataset.
> We have addressed the ambiguity in lines 248-252,regarding how student views and teacher views are created. The new changes are reflected in the updated manuscript
>
>
> To address the concerns regarding the impact of embedding misalignment and its effect on downstream task performance, we have clarified the concept of alignment in the general comments section "CLIP Alignment Redefined." Additionally provide visualizations in the Appendix Section A.3 which showcases how improved alignment leads to consistent gains across tasks, even when numerical improvements may appear modest.

---

> > ### Comment · Reviewer_nEF5 · 2024-11-26
> >
> > Thanks for the author's reply. I still believe that the unfair comparison with baseline methods is a big problem, so I decide to remain my score.
> >
> > I hope the author can introduce more baseline methods that utilize all unlabeled data for comparison to demonstrate the method's effectiveness.

---

### Official Review · Reviewer_M2x7 · 2024-11-03

**Soundness:** 2
**Presentation:** 1
**Contribution:** 2
**Rating:** 3
**Confidence:** 3

**Summary:**

The paper introduces CLIP-Enhance, a method to improve CLIP's zero-shot classification by addressing two key issues: misalignment between text and image embeddings and the long-tailed distribution of the training data. CLIP-Enhance reinterprets zero-shot classification as a clustering problem on a hypersphere using a von Mises-Fisher (vMF) mixture model. This model optimizes both the alignment and concentration of embeddings through self-supervised learning, inspired by the DINO framework. The empirical findings reveal a remarkable enhancement in the performance of zero-sample classification across multiple benchmarks, manifesting superior management of data distribution disparities and robustness under limited data circumstances.

**Strengths:**

1.As claimed by the authors, they propose a SOTA zero-shot classification structure based on CLIP, which performs well on related benchmarks and limited compute/data resources.

2.The novel vMF mixture model proposed in the paper is justified, as the knowledge distillation introduced in the confidence selection section is quite reasonable, and the convergence of von Mises-Fisher clustering can be mathematically substantiated.

**Weaknesses:**

1.More ablation experiments are needed to increase confidence of your method, as experimental result comparisons with counterparts on some important datasets are missing.

2.There are some imprecise statements in the paper, and the CLIP-Enhance method lacks visual demonstration of results. It is necessary to add the relevant content in the appendix.

Please see the questions section for more details.

**Questions:**

1.Your experiments on 10 datasets are quite far from enough. As your most competitive counterparts, CALIP and TPT have both conduct experiments on ImageNet-V2/ A / R/ Sketch, as well as other smaller but typical datasets like OxfordPets and StanfordCars. I have no idea why your work ignore those important experiments. Maybe lack of time for the submission deadline? If so, you can supplement the information during the subsequent rebuttal phase. Once the requirements are met, I will reset my review score.

2.Using only naive TPT as the baseline in your ablation studies is unreasonable. At minimum, the state-of-the-art (SOTA) results from CoOp+TPT should also be included as a baseline. As we all know, CoOp+TPT outperforms naive TPT across the board, and for CLIP-Enhance to be considered a SOTA method, it needs to surpass those results of CoOp+TPT.

3.The statement of 'This is an image of a [CLASS].' isn't accurate, as the default use of CLIP prompt template is 'A photo of a [CLASS].'. Like  the figure 1 presented in your paper, these two prompt templates are different in embedding hypersphere and bring discrepancies to the experimental results, even if the difference is quite slight. Scientific writing should be professional and rigorous, and 'the devil is in the details'. You may have been careless, but I still can't believe that such a rookie mistake would appear in a paper submitted to ICLR. I hope you can carefully check the references and mathematical derivations in the paper again, to ensure the accuracy and standardization of the article.

4.More experimental details and visualized experimental results can be placed in the appendix, which would further analyze and demonstrate your method, allowing readers to more intuitively understand the value and significance of your work. Although ICLR has restrictions on the length of the main text, detailed mathematical derivations and experimental results can be elaborated in the appendix section.

---

> ### Author Response · Authors · 2024-11-25
>
> We have conducted additional experiments on Caltech101, Stanford Cars, Oxford-IIIT Pets, UCF101, and ImageNet-Sketch. These results, included in the updated manuscript, provide a more comprehensive comparison with state-of-the-art methods.
>
> CoOp is a few-shot learning method that relies on image-text labeled pairs, making it incompatible with the zero-shot classification protocol followed in our work. To maintain consistency with the zero-shot paradigm, we chose naive TPT as our baseline.
>
> We have also added detailed visual demonstrations and further analyses in the appendix. These additions provide deeper insights into the impact of embedding alignment on downstream performance. Furthermore, we have elaborated on the rationale behind our design choices in a new section, "CLIP Alignment Redefined," included in the general comments and appendix.

---

> > ### Comment · Reviewer_M2x7 · 2024-11-25
> >
> > Thanks for your response. I remain my score.

---

### Official Review · Reviewer_jrhq · 2024-11-04

**Soundness:** 3
**Presentation:** 4
**Contribution:** 3
**Rating:** 5
**Confidence:** 4

**Summary:**

This paper proposes a vMF-based teacher-student knowledge distillation tuning framework to enhance the CLIP zero-shot performance. Technically, it proposes two strategies: (1) closed-form approximation of normalization term of vMF distribution to model variance statistics of different categories; (2) knowledge distillation with confident view selection for stabler fine-tuning. The experimental results demonstrate the superior performance of CLIP-Enhance compared with some baselines on multiple downstream zero-shot classification benchmarks.

**Strengths:**

Strengths:

- The idea of closed-form approximation of the vMF normalization constant is novel.
- The ablation results can validate the effectiveness of vMF distribution modeling.

**Weaknesses:**

Limitations:

- Missing baselines: It seems there is a line of missing related works that are not compared, including [1], [2], [3], [4].
- Insufficient experiments: (1) Although the author claims that the proposed method with vMF distribution can address the long-tail issue of CLIP, it is still indispensable to ablate the long-tailed downstream tasks. (2) What are the performance gains from the confident view selection? Why it is top 10%?
- Question: How does the learned $\kappa$ differ from different classes? Better to present statistics or analysis. Why cannot tune the existing blocks of CLIP and add an additional projector?
- The alignment calibration between visual and language embeddings is minor. Could you please provide more results on additional baselines to get the knowledge of the difficulty of this task? This can better solidify your hypothesis on the explanatory factors.

[1] PLOT: PROMPT LEARNING WITH OPTIMAL TRANSPORT FOR VISION-LANGUAGE MODELS

[2] MaPLe: Multi-modal Prompt Learning

[3] AWT: Transferring Vision-Language Models via Augmentation, Weighting, and Transportation

[4] Mixture of Prompt Learning for Vision Language Models

**Questions:**

All my concerns are listed in the weaknesses.

---

> ### Author Response · Authors · 2024-11-24
> **Response**
>
> (Missing Baselines): We acknowledge the relevance of the cited works. However, the methods in [1], [2], [3], and [4] either incorporate labeled image-text pairs or leverage subsystems like CoOp, a few-shot learning approach. These methodologies deviate from the zero-shot evaluation regime we target, making direct comparisons with them less appropriate.
> We provide further explanation about how our model addresses both alignment and issues caused by the long tail distribution of CLIPs training data in general comments section "CLIP Alignment Redefined." While the alignment calibration improvements may appear minor, we emphasize that they align with the inherent difficulty of balancing visual and textual embeddings in zero-shot settings
>
> The choice to choose top 10% was inspired by the success achieved by TPT, which employs a similar evaluation protocol, Furthermore we have updated the ablation section of our paper with how ensemble eval affects the classification problem.
>
> We present statistical analyses on the learned κ\kappaκ values across different classes in Appendix A.2 and Appendix A.3, providing insight into its variability and contribution to performance
>
>
> These revisions aim to comprehensively address your concerns and provide a stronger foundation for our contributions.

---

> > ### Comment · Reviewer_jrhq · 2024-11-25
> >
> > I appreciate the efforts of the authors to clarify the main setting of this work. Initially, the \textit{zero-shot} is a little confusing since there have been various different settings under this category. To clarify, the \textit{zero-shot} in this work denotes the unsupervised adaptation of CLIP to test data for classification tasks without fine-tuning on some seen classes, e.g., CoOp. However, the authors still seem to miss an important related work that achieves the same goal [1]. I consider it necessary to have a direct comparison. As mentioned by reviewer EmTj, it is less proper to compare with test-time tuning methods due to fairness issues. Besides, when authors claim the parameter efficiency compared with previous tuning methods, it is also necessary to list the FLOPs or #params or wall-clock tuning time to clarify, since this work also involves dataset-specific tuning at test time. Better clarification in related work is required.
> >
> > Taking into consideration the above points, in addition to quality of presentation, I would like to maintain my score.
> >
> > References:
> >
> > [1] Masked Unsupervised Self-training for Zero-shot Image Classification

---

> > ### Comment · Reviewer_jrhq · 2024-11-26
> >
> > My concerns about the hyperparameters have been addressed.

---

### Official Review · Reviewer_jQ4w · 2024-11-04

**Soundness:** 2
**Presentation:** 3
**Contribution:** 2
**Rating:** 3
**Confidence:** 3

**Summary:**

This paper propose a method named CLIP-Enhance to improve CLIP zero-shot classification. CLIP-Enhance addresses two challenges: (1) the misalignment between image and text embeddings, and (2) the long-tailed distribution of CLIP's training data. The authors re-fomulate zero-shot classification as a von Mises-Fisher clustering problem. Then an self-supervised learning method is proposed to optimize both alignment and concentration. Experiment results show that the proposed method can improve zero-shot classification accuracy.

**Strengths:**

1. The idea of reformulating CLIP zero-shot classification as a clustering problem is interesting.
2. This paper is well-organized and easy to follow.

**Weaknesses:**

1. My major concern is the limited evaluation. The results of CALIP and TPT on C-10, C-100 and MNIST are not provided. The results on Caltech101, Aircraft, Stanford Cars and UCF101 are not provided, which are widely used for zero-shot evaluation in prior works.
2. The article claims to improve the misalignment between class embedding and image embedding, but how does this misalignment affect downstream task performance? The article needs to provide a more in-depth analysis.
3. The reported improvement of "alignment" in Table 2 is too small to support the conclusion. Have the authors considered using other metrics or methods to demonstrate it? such as visualizations of the embedding space or quantitative measures of clustering quality.
4. The sensitivity of the initial concentration parameter was briefly examined. But when $\kappa_0 = 5000$, the performance drop significantly on F-102 (67.3 $\rightarrow$ 11.3) and MNIST (64.95 $\rightarrow$ 53.1). Further analysis of the robustness of the proposed method on different $\kappa_0$ is needed.

**Questions:**

see weaknesses

---

> ### Author Response · Authors · 2024-11-24
> **Response**
>
> Limited Evaluation : . In response, we have incorporated results on Caltech101, Stanford Cars, OxfordIIITpets and UCF101as well as Imagenet-Sketch. These additions are detailed in the updated manuscript to ensure a comprehensive comparison with prior works.
>
> Impact of Misalignment and Small Alignment Improvement : To address the concerns regarding the impact of embedding misalignment and its effect on downstream task performance, we have clarified the concept of alignment in the general comments section "CLIP Alignment Redefined." Additionally provide visualizations in the Appendix Section A.3 which showcases how improved alignment leads to consistent gains across tasks, even when numerical improvements may appear modest.
>
> Sensitivity to Initial Concentration Parameter: The performance degradation at high κ0\kappa_0κ0​ values (e.g., κ0=5000\kappa_0 = 5000κ0​=5000) stems from numerical instability in our method. This is now thoroughly analyzed in the updated ablation section (A.2), where we also explore the robustness of our approach across a range of κ0\kappa_0κ0​ values. Our findings suggest practical bounds for κ0\kappa_0κ0​ to mitigate instability while maintaining competitive performance.

---

> ### Comment · Reviewer_jQ4w · 2024-11-26
> **Response**
>
> Thanks for your response and additional results. However, there are still some 'N/A' results in Table 1. And the baseline methods are still limited and not appropriate enough.
>
> I will keep my original rating.

---

### Author Response · Authors · 2024-11-22
**Answering common concerrns raised by the reviewers**

## Updated Zero-Shot Baseline Comparison
In response to feedback from multiple reviewers, we have expanded our zero-shot baseline comparison to include additional fine-grained classification datasets. Table 1 in our paper has been updated to reflect these results, covering the following datasets:
- **OxfordIIITPets**: A dataset with classes from cat and dog families.
- **Stanford Cars**: A dataset for classifying various car models.
- **Caltech101**: A dataset with 101 object categories.
- **UCF101**: A dataset focused on classifying human actions.

## Analyzing Kappa initialization
we provide results on the impact of different choices for the initial concentration parameter κ0. Results from κ0 = 500 and κ0 = 5000 are provided in Table 6 along with the results of the value we used everywhere else in the paper, namely κ0 = 3000. Building on the discussion in § 4.3 regarding the choice of κ0, we further illustrate in Figure 3 how increasing κ0 reduces the contribution of the normalization term to the logit score. This observation provides insight into the observed performance degradation: as the model increasingly relies on the normalization component rather than the image data for prediction its performance declines, around the κ0 = 2000 the contribution reaches close to zero and we observe that model accuracy drop near the values obtained by our model without Vmf formulation as discussed in section § 5.2. Additionally, we observed that larger values of κ0 result in numerical instability during training in our setup, leading to a significant drop in performance.

## CLIP alignment Redefined
CLIP's training objective aims at a relative alignment of captions embeddings with embeddings of their corresponding image embeddings [1]. More precisely, for a random batch from CLIP's training dataset, the symmetric cross entropy is minimized when 1): every caption embedding is more are aligned (larger cosine similarity) with its corresponding image embedding than with any other image embedding 2) every image embedding is more are aligned (larger cosine similarity) with its corresponding caption embedding (positive pair) than with any other caption embedding (negative pairs). Moreover, since “a temperature parameter which controls the range of the logits in the softmax is directly optimized during training” -- which would appear to take approximately the value $\exp(\tau)=100$ [2], which is the maximal value allowed during training to prevent instability [1] -- a small difference between the cosine similarity of a positive pair compared to negative pairs leads to cross entropy loss which is roughly zero. For instance, for the training batch size ($b=32,768$), a positive cosine similarity of $0.5= \cos(60^\circ)$ for negative cosine similarities of $0.225\approx \cos(77^\circ)$ obtains a cross entropy loss (when logits are scaled by $100$) which is smaller than the smallest representable number (with Pytorch, Float32).

Hence, CLIP's training does not aim at aligning text embeddings and their corresponding images in the sense that the angle between them is small (cosine almost $1$). On the contrary, in CLIP's multimodal embedding space, the angle between a text representation and its corresponding image representation can be as large as $60^\circ$ (cosine $0.5$) as long as negative pairs are less aligned, namely with an angle larger than $77^\circ$. Therefore, we conclude that the quality of CLIP's multimodal embeddings really resides in their RELATIVE alignment, not in the actual alignment of positive pairs, an observation that is also supported by [3].

Furthermore, we consider that this relative alignment highly depends on the training distribution. When leveraging these representations for a downstream classification task, we aim at a relative alignment of text-based class reference embeddings with respect to the image distribution. A specific balanced classification task like CIFAR10 is expected to differ greatly from the long tail distribution of CLIP's training dataset, resulting in a poor relative alignment. Our approach aims at adapting in an unsupervised manner the initial text-based class reference embeddings to improve their relative alignment on the image dataset. By endowing each class reference with a concentration parameter and formulating the classification problem as a von Mises--Fisher mixture model, our approach achieves this goal as reflected by the improved accuracy.

[1] Radford, Alec, et al. "Learning transferable visual models from natural language supervision." International conference on machine learning. PMLR, 2021.

[2] CLIP's official git repository. API. url:https://github.com/openai/CLIP.

[3] Liang, Victor Weixin, et al. "Mind the gap: Understanding the modality gap in multi-modal contrastive representation learning." Advances in Neural Information Processing Systems 35 (2022): 17612-17625.

We have updated the paper to better explain the questions raised by the reviewers

---

### Comment · Area_Chair_dGCA · 2024-11-25
**Interactive Discussions**

Dear Reviewers,

Thank you for your efforts in reviewing this paper. We highly encourage you to participate in interactive discussions with the authors before November 26, fostering a more dynamic exchange of ideas rather than a one-sided rebuttal.

Please feel free to share your thoughts and engage with the authors at your earliest convenience.

Thank you for your collaboration.

Best regards, ICLR 2025 Area Chair

---

### Meta-Review · Area_Chair_dGCA · 2024-12-19

**Metareview:**

This paper explores clustering in unsupervised (zero-shot) classification using CLIP and introduces a vMF-based teacher-student knowledge distillation framework. While the clustering approach shows potential, the work suffers from notable experimental shortcomings, as highlighted by the reviewers. These include incomplete results (e.g., NA in Table 1), missing key baselines, and limited dataset coverage. Additionally, reviewers flagged the term "zero-shot classification" as potentially inappropriate, suggesting "unsupervised classification" as more accurate. Following the author-reviewer discussion, all reviewers recommended rejection. The Area Chair concurred with their assessment and also recommended rejection.

**Additional Comments On Reviewer Discussion:**

A common concern is the lack of comprehensive comparisons with previous methods. The authors should consider incorporating additional comparisons across both datasets and unsupervised classification baselines in future submissions. Furthermore, the distinctions between the proposed method and prior work, such as TPT, as well as self-supervised approaches like SimCLR, should be articulated more clearly.

---

### Decision · Program_Chairs · 2025-01-22

Reject